# Human Papillomavirus Types and Cervical Cancer Screening among Female Sex Workers in Cameroon

**DOI:** 10.3390/cancers16020243

**Published:** 2024-01-05

**Authors:** Simon M. Manga, Yuanfan Ye, Kathleen L. Nulah, Florence Manjuh, Joel Fokom-Domgue, Isabel Scarinci, Alan N. Tita

**Affiliations:** 1Women’s Health Program, Cameroon Baptist Convention Health Services, Bamenda P.O. Box 1, Cameroon; nukaleps@gmail.com (K.L.N.); manjuhrencefav@yahoo.com (F.M.); fokom.domgue@gmail.com (J.F.-D.); 2Center for Women’s Reproductive Health, Department of Obstetrics & Gynecology, University of Alabama at Birmingham, 1700 6th Avenue South, Suite 10270, Birmingham, AL 35233, USA; cynthiay@uab.edu (Y.Y.); iscarinci@uabmc.edu (I.S.); atita@uab.edu (A.N.T.); 3Division of Cancer Prevention and Population Sciences, The University of Texas MD Anderson Cancer Center, 1155 Presser Street, Houston, TX 77030, USA

**Keywords:** human papillomavirus, female sex workers, genotype, thermal ablation, large loop excision of the transformation zone, Cameroon

## Abstract

**Simple Summary:**

Female sex workers (FSWs) are exposed to sexually transmitted infections (STIs) due to frequent indiscriminate sexual intercourse with several persons. The human papillomavirus (HPV), the principal cause of cervical cancer, is among the commonest STIs worldwide. It is thought that FSWs have a very high prevalence of HPV and cervical precancers due to this occupational exposure. This study was conducted to investigate the prevalence of HPV, HPV types, and cervical precancers among FSWs in Cameroon. The results of this study will highlight the importance of targeting vulnerable populations for cervical cancer prevention to facilitate the attainment of the WHO worldwide cervical cancer elimination strategy by 2030.

**Abstract:**

Background: Female sex workers (FSWs) are at high risk for sexually transmitted infections (STIs), including infection with human papillomavirus (HPV) and cervical cancer due to occupational exposure. The objective of this study was to estimate the prevalence of HPV, HPV types, and precancerous lesions of the cervix among FSWs in Cameroon. Material and Methods: In this cross-sectional study, FSWs in Cameroon aged 30 years and above were screened for cervical cancer using high-risk HPV testing and genotyping and visual inspection with acetic acid and Lugol’s iodine (VIA/VILI) enhanced using digital cervicography (DC) simultaneously. Those who were positive for VIA/VILI-DC were provided treatment with thermal ablation (TA) immediately for cryotherapy/TA-eligible lesions while lesions meeting the criteria for large loop excision of the transformation zone (LLETZ) were scheduled at an appropriate facility for the LLETZ procedure. HPV-positive FSWs without any visible lesion on VIA/VILI-DC were administered TA. Bivariate analyses were conducted to compare demographic and clinical characteristics. Crude and adjusted logistic regression models were computed for HPV infection status and treatment uptake as outcomes in separate models and their ORs and 95% confidence intervals (95% CI) were reported. Results: Among the 599 FSWs aged 30 years and older that were screened for HPV and VIA/VILI-DC, 62.1% (95% CI: (0.58–0.66)) were positive for one or more HPV types. HPV type 51 had the highest prevalence (14%), followed by types 53 (12.4%) and 52 (12.2%). Type 18 had the lowest prevalence of 2.8% followed by type 16 with 5.2%. In the multivariable model, HIV-positive FSWs were 1.65 times more likely to be infected with HPV compared to their HIV-negative counterparts (AOR: 1.65, CI: 1.11–2.45). A total of 9.9% of the 599 FSWs were positive for VIA/VILI-DC. Conclusion: The prevalence of HPV infection among FSWs in Cameroon is higher than the worldwide pooled FSW prevalence. HPV types 51 and 53 were the most prevalent, while types 18 and 16 were the least prevalent. HIV status was the only variable that was significantly associated with infection with HPV.

## 1. Introduction

Due to occupational exposure, female sex workers (FSWs) are at high risk for sexually transmitted infections (STIs) including human papillomavirus (HPV), which is the principal cause of cervical precancer and cancer [1,2]. HPVs are subdivided into two groups: the low-risk types including HPV types 1, 2, 6, 8, 11, 34, 40, 42, 43, 44, 61, 69, 71, 72, 81, 83, and 84, which are commonly associated with benign manifestations such as warts, papillomas, condylomata, and focal epithelial hyperplasia; and high-risk types including HPV types 16, 18, 31, 33, 35, 39, 45, 51, 52, 53, 56, 58, 59, 66, 68, 73, and 82 which are associated with malignant manifestations [3]. The cervical precancerous lesion is asymptomatic and takes several years to progress to invasive cervical cancer (ICC) [1,4]. The precancerous lesion can be treated to prevent its progression into ICC. Treatment for cervical precancerous lesions is very effective with a cure rate of almost 100% [5]. In addition to cervical cancer, it has been established that persistent infection with high-risk HPVs is also associated with anogenital as well as head and neck cancers [6,7,8]. Furthermore, the risk of acquiring human immunodeficiency virus (HIV) is higher among FSWs compared to the general population. The viral proteins gp120 and tat and the HIV-associated activation of inflammatory processes in the mucosal epithelium can lead to the disruption of epithelial junctions, which may promote the penetration and/or dissemination of other viruses, including herpes viruses (HSVs) and HPVs [9]. One of the key events of HPV-induced carcinogenesis is the integration of the HPV genome into a host chromosome [10].

The FSW population in Cameroon is estimated to be 112,580 or about 2% of females aged 15–64 years. FSWs in Cameroon were reported to have an HIV prevalence of 36.5% compared to a national prevalence of 3.4% among women [11]. The prevalence of cervical cancer is about three to five times higher among women infected with HIV compared to their HIV-negative counterparts [1,12]. Given the high prevalence of HIV among FSWs, they are at an even greater risk of developing cervical cancer. Due to their perceived high HPV prevalence, FSWs are thought to play an important role in the burden of HPV and its associated diseases [13].

FSWs may be less likely to seek cancer screening because they have to pay out-of-pocket to receive screening services since health insurance or government coverage does not exist in Cameroon [11]. Most FSWs are of low socio-economic status (SES); thus, they may end up seeking healthcare services only when they are in an acute health crisis. In the case of cervical cancer, they are likely to present at advanced stages of the disease when treatment is unaffordable and less effective. Moreover, cancer treatment in Cameroon, especially for cervical cancer which may require radical surgery and/or radiation or chemoradiation is very expensive and not optimal [11]. Therefore, screening and treatment of pre-invasive lesions to reduce the morbidity and mortality among this vulnerable group are paramount. 

HPV DNA testing which identifies the presence of high-risk or oncogenic HPVs on the cervix has become the most effective test for cervical cancer screening [14]. In 2021, the World Health Organization (WHO) recommended HPV DNA as the preferred screening strategy for cervical cancer in all settings [15]. The WHO has proposed two HPV screening approaches for women aged 30 years and over (or 25 years and over for those living with HIV); the “HPV see-and-treat approach”, and the “HPV see-triage-and-treat approach”. In the HPV see-and-treat approach, all women positive for oncogenic HPV are treated. Visual inspection with acetic acid (VIA) is first conducted to determine the presence or absence of a precancerous or cancerous lesion. Those without any visible lesions on VIA are treated with ablative treatment such as cryotherapy or thermal ablation (TA) while those with visible lesions are treated using ablation or excision (large loop excision of the transformation zone (LLETZ)) depending on the characteristics of the lesion. For the HPV see-triage-and-treat approach, every positive HPV test is triaged with another test, VIA or cytology, and the treatment depends on the results of the triage test. If the triage test is positive, she is treated; otherwise, she is not treated. The HPV see-and-treat approach appears to be more appropriate for high-risk and vulnerable populations like FSWs because getting them for triage and follow-up can be quite challenging.

The objective of this study was to determine the prevalence of high-risk HPVs, high-risk HPV types, and precancerous lesions of the cervix among FSWs in Cameroon. 

## 2. Populations and Methods

### 2.1. Study Population, Setting, and Procedure 

#### 2.1.1. The Study Type

We conducted a cross-sectional study whereby female sex workers (FSWs) in Cameroon aged 30 years and above were screened for cervical cancer using human papillomavirus (HPV) genotyping and visual inspection with acetic acid and Lugol’s iodine (VIA/VILI) enhanced using digital cervicography (DC) simultaneously. 

The study received Institutional Review Board (IRB) approval from the Cameroon Baptist Convention Health Services (CBCHS) and the University of Alabama at Birmingham (UAB). Participants signed an informed consent and the data collected complied with relevant patient data protection rules and guidelines.

The project ran from January to October in the year 2020. The HPV genotyping was performed using an AmpFire HPV analyzer, Atila BioSystems, Sunnyvale, CA, USA. The AmpFire technology uses multiplex isothermal real-time fluorescence to detect HPV in cervico-vaginal specimens and it can genotype 15 high-risk types of HPV including types 16, 18, 31, 33, 35, 39, 45, 51, 52, 53, 56, 58, 59, 66, and 68 [16]. 

#### 2.1.2. The Setting

The screening was performed by nurses of the women’s health program (WHP) of the CBCHS. The CBCHS is a large non-profit, faith-based healthcare organization with a network of 95 health facilities located in eight of the ten regions of Cameroon. The WHP is a women’s reproductive health program that has integrated cervical cancer prevention, breast cancer screening, family planning, and syndromic management of reproductive tract infections. The WHP is the most comprehensive cervical cancer prevention program in Cameroon and the Central African sub-region [17]. The WHP screens an average of 8000 women per year for cervical cancer and has screened over 120,000 women. For women 25 to 29 years old, screening is performed using VIA/VILI-DC, while for women 30 years and older, screening is performed with the AmpFire HPV analyzer, and those positive for one or more types of HPV are triaged to VIA/VILI-DC. In DC, a camera is used to take real-time highly magnified images of the cervix projected onto a TV monitor so that both the woman and the provider can see the cervix life at the same time, mimicking video colposcopy [18]. Precancerous lesions of the cervix that meet cryotherapy eligibility are treated with cryotherapy or thermal ablation (TA) while lesions that meet eligibility for large loop excision of the transformation zone (LLETZ) are treated with LLETZ. A biopsy is performed on lesions suspicious of invasive cervical cancer (ICC) and if the histopathology confirms the ICC, the patient is referred to a specialized center for appropriate treatment.

For successful implementation, the WHP collaborated with Horizons Femmes (HF), a non-governmental organization (NGO) in Cameroon that promotes and protects the health of FSWs in three major cities of Cameroon (Yaoundé, Douala, and Bafoussam). The main activities of HF are the provision of HIV testing and navigation of those positive into HIV care and treatment which is run by the NGO. Those who are HIV-negative are followed up every three months for HIV re-testing. The HF clinics are located within the hottest spots for sex work in these mentioned cities. Each clinic has a laboratory with technicians to run HIV testing and viral loads as well as other tests required for initiation and follow-up of persons on antiretroviral (ARV) drugs. They have a pharmacy that supplies ARVs to FSWs living with HIV. They have psycho-social workers and nurses to attend to the basic health needs of the FSWs including counseling and dispensing male and female condoms. When a FSW within their network develops a complication from ARVs or needs further evaluation, a staff member of the NGO accompanies her to the hospital in what is known as an active referral. 

HF has trained and hired experienced FSWs as peer educators whose role is to go out with sensitization messages on the importance of HIV testing, adherence to ARVs, and the correct and consistent use of condoms. They also mobilize the FSWs whenever the NGO needs them for group education or other activities. For this project, 25 peer educators were trained as follows; 10 in Yaoundé, 10 in Douala, and 5 in Bafoussam. The one-day onsite training was focused on the basics of STIs/HPV and cervical cancer. Brochures and posters were developed and were handed to these peer educators to facilitate the mobilization of their peers. A referral card was developed and handed to the peer educators and every FSW referred for screening was provided a card that carried her name and the name of the peer educator referring her. Each peer educator was entitled to USD 1 per referral.

#### 2.1.3. Inclusion and Exclusion Criteria

Inclusion criteria: The main inclusion criteria were women 30 years and above who self-identify themselves as FSWs.Exclusion criteria: Our exclusion criteria included (1) pregnancy, (2) previous total abdominal hysterectomy, and (3) women with already diagnosed invasive cervical cancer.

#### 2.1.4. Data Collection

The routine WHP questionnaire for cervical cancer screening was used to collect data for this study. The questionnaire is divided into two main parts, demographic and medical history and physical exams. For this study, an additional supplemental questionnaire was developed and used. The supplemental questionnaire captured data on risk behaviors that pertained to sex work. The supplemental questionnaire and the demographic and medical history part of the WHP questionnaire were completed by the social workers of HF. The WHP nurses then completed the portion for physical exams at the end of the screening.

#### 2.1.5. The Procedure

The specimens for HPV were collected by the nurses using the AmpFire sterile brush. The long brush was inserted into the participant’s vagina after slightly separating the labia. The brush was inserted until resistance was met, then it was rotated three times and removed. The brush was then cut from the brush holder and the brush deposited into a specimen container, locked, labeled, and shipped to the lab. The specimens from Bafoussam and Yaoundé were analyzed at EtougEbe Baptist Hospital in Yaoundé while those from Douala were analyzed at the Mboppi Baptist Hospital in Douala. The HPV results were available the following day. 

Immediately after the collection of the specimen for HPV, a plastic disposable bi-valve vaginal speculum was inserted into the participant’s vagina for VIA/VILI-DC. Those who were positive for VIA/VILI-DC were provided treatment with TA immediately for cryotherapy/TA-eligible lesions while lesions meeting the criteria for LLETZ were scheduled at the end of the screening week in Bafoussam Baptist Hospital for those in Bafoussam, EtougEbe Baptist Hospital Yaoundé for those in Yaoundé, and Mboppi Baptist Hospital Douala for those in Douala. All three CBCHS hospitals are a taxi drop away from the HF center in the three cities. Participants for LLETZ were accompanied to the CBCHS hospital by the staff of HF as part of the active referral system. All LLETZs were performed by the principal investigator (PI). For those positive only for HPV, without any visible lesion on VIA/VILI-DC, TA was provided to them upon collection of their results which was typically the following day. It should be noted that, during the project, we screened FSWs aged 21 to 29 with VIA/VILI-DC only, but their data have not been included in this study.

### 2.2. Data Analysis

Bivariate analyses were conducted to compare baseline demographic and clinical characteristics between HPV infection status, VIA/VILI-DC status, and treatment uptake separately. All variables were grouped into categorical variables, and Chi-square tests were used to compare differences between them. Crude logistic regression models were fitted using individual baseline characteristics as a single predictor and HPV infection status and treatment uptake as outcomes in separate models. Significant baseline characteristics from the bivariate analysis were added to multivariable logistic regression models. Both crude odds ratios (ORs) and adjusted ORs along with their 95% confidence intervals (95% CI) were calculated. In addition to examining the factors associated with the three outcomes mentioned above, we also plotted a bar chart based on the estimated prevalence of 15 HPV subtypes among all participants. Statistical analyses were performed using SAS 9.4 (SAS Institute, Cary, NC, USA).

## 3. Results

We screened a total of 599 female sex workers (FSWs) aged 30 years and older for both human papillomavirus (HPV) genotyping and visual inspection with acetic acid and Lugol’s iodine (VIA/VILI) enhanced using digital cervicography (DC). Of the 599 FSWs, 37 (6.2%) were positive for both HPV and VIA/VILI-DC, 322 (53.8%) were positive for HPV and negative for VIA/VILI-DC, 20 (3.3%) were HPV-negative and VIA/VILI-DC-positive, while 196 (32.7%) were both HPVnegative and VIA/VILI-DCnegative (Figure 1).

### 3.1. HPV Genotyping Results

Among the 599 records, a total of 62.1% (95%CI: 58.2–66.0%) (Table 1) were positive for one or more HPV types. All 15 HPV types detected using the AmpFire technology were present in this population. Among the 15 HPV types, type 51 had the highest prevalence of 14%, followed by type 53 (12.4%) and type 52 (12.2%). Type 18 had the lowest prevalence of 2.8% followed by type 16 with 5.2% (Figure 2). 

The distribution of site of screening, HIV status, monthly income, and condom use was significantly different between the HPV-positive and HPV-negative groups (Table 1). The multivariable logistic regression model on predictors for infection with HPV included age, marital status, other occupations, educational level, duration of sex work, religion, income generated, condom use, HIV status, and VIA/VILI-DC status (Table 2). FSWs living with HIV were 1.65 times more likely to be infected with HPV compared to their HIV-negative counterparts (AOR: 1.65, 95% CI: 1.11–2.45). Site of screening, monthly income, and condom use lost their significance in the multivariable regression model.

### 3.2. VIA/VILI-DC Results

Table 3 shows participants’ characteristics according to VIA/VILI-DC results. A total of 57 FSWs (9.9%) were positive for VIA/VILI-DC. Only site of screening (*p* = 0.035) and HIV status (*p* = 0.033) showed significance in the bivariate model. 

### 3.3. Treatment Uptake

FSWs positive for VIA/VILI-DC were provided treatment according to the characteristics of the lesion. Smaller lesions were treated with thermal ablation (TA) while bigger lesions were treated with large loop excision of the transformation zone (LLETZ). Those who were HPV-positive but had no VIA/VILI-DC lesions were provided TA. Among the 37 FSWs positive for both HPV and VIA/VILI-DC, 10 (27%) were treated with LLETZ, and 20 (54.1%) were treated with TA (Figure 1). Of the 322 FSWs positive for HPV and negative for VIA/VILI-DC, 123 (38.2%) were treated with TA. Of the 20 FSWs who were HPV-negative and VIA/VILI-DC-positive, 11 (55%) were treated with TA. Of the 14 FSWs who were eligible for LLETZ, 10 (71.4%) were treated. Among the 10 LLETZ, the histopathology report for 9 (90%) returned as CIN2+, while 1 (10%) returned as CIN1 (Figure 1). Among the FSWs who were eligible for treatment, among those who received treatment, and those who did not receive treatment, only HIV status (*p* = 0.03) and site of screening (*p* = 0.002) were significant in the bivariate model (Table 4). However, they both lost their significance in the multivariable model (Table 5). 

## 4. Discussion

### 4.1. Human Papillomavirus Genotyping

We found a high prevalence of human papillomavirus (HPV) infection of 62.1% among female sex workers (FSWs) in Cameroon. Our prevalence in this population was higher than the reported prevalence of FSWs in several countries ranging from 5.5 to 57.7% [13,19,20,21,22,23,24,25]. However, our prevalence was less than the 79.8% in Senegal [26] and the 66.8% in Peru [27]. All these studies including those from Senegal and Peru recruited FSWs as young as 18 and 15 years. Whereas our study recruited FSWs from the age of 30 years. It is well known that transient HPV infection is higher among women younger than 30 years which is the reason why the WHO recommends screening for HPV to begin at age 30 [15]. Our prevalence of 62.1% was higher than the pooled prevalence of 42.6% CI: 38.5 to 46.7 from a meta-analysis of HPV studies among FSWs worldwide [28]. 

In this study, only HIV status was significantly associated with HPV infection. FSWs living with HIV were more likely to be infected with HPV compared to their HIV-negative counterparts. This was similar to what was found in Togo where 15.7% of HIV-positive FSWs were positive for high-risk HPV and 8.2% were negative (*p* = 0.004) [22]. In a 20-year systematic literature review of HPV prevalence among HIV-positive and HIV-negative women in the general population of Sub-Saharan Africa (SSA), the prevalence of various HPV types and multiple HPV infections was higher in HIV-positive women compared to HIV-negative women (*p* < 0.001) [29]. A study conducted in Togo showed that the HIV-positive FSWs predisposed to higher HPV infection were those with inadequate HIV virologic control [30]. A large meta-analytic systematic review of worldwide longitudinal studies on HPV incidence and clearance rate by HIV status and on HIV incidence by HPV status has demonstrated that a lower CD4 count is associated with a high prevalence of HPV infection and disease [31].

In our study, condom use was not significantly associated with infection with HPV. It is common knowledge that the correct and consistent use of condoms prevents sexually transmitted infections (STIs) including HPV [32]. However, the protection of condoms against HPV is less effective compared to other STIs. This is because persons infected with HPV might carry the infection on their pubis and thighs, parts of the body that cannot be covered by the condom during sexual activity [33]. Just like our study, several studies did not find any statistically significant association between condom use and HPV infection among FSWs [13,20,21,22,34]. On the contrary, a study conducted in Benin and Mali found that condoms were able to protect against infection with HPV among FSWs (*p* = 0.004) [35].

The income generated was not statistically significantly associated with infection with HPV. FSWs who generate more income are more likely to be engaged in unprotected sex at higher charge and they might be more vulnerable to HPV infection. We did not find any studies that have examined the relationship between sex work income and HPV infection.

Though all 15 types of high-risk HPV in the AmpFire platform were present in our FSWs population, the most prevalent were types 51, 53, and 52, while types 18 and 16 were the least prevalent. In a systematic literature review and meta-analysis of studies performed on HPV prevalence among FSWs in SSA, type 16 was the most prevalent followed by types 52 and 53 [27]. In a South African study among HIV-positive and HIV-negative women, HPV type 16 was the most prevalent [35]. Though types 52 and 53 were among the top three types in this study, type 16, which came in the first position, was the least prevalent in our population. In a large prospective study on HPV genotyping among women in the general population in Mexico that ran for 22 years, enrolled over 7000 women, and had an overall prevalence of high-risk HPV of 54.17%, HPV 16 was the most prevalent type [36]. The reason for the low prevalence of HPV 16 in our population is not known. The only HPV vaccine available in Cameroon is Gardasil-4, a recombinant HPV quadrivalent type 6, 11, 16, and 18, vaccine [37]. If HPV 16 and 18 are the least prevalent high-risk HPV types in Cameroon, then the only available HPV vaccine might not be of optimal public health benefit in the country Our team published a case report from Cameroon about a 26 years old lady who developed four anogenital cancers at the same time including cervical cancer, vaginal cancer, vulvar cancer, and anal cancer. Both HPV 16 and 18 were absent in her genotype. Instead, her genotypes were 33, 51, and 68 [8]. This is a pointer that HPV types other than types 16 and 18 may be more virulent in Cameroon. Even though this lady had no history of HPV vaccination, if she had been vaccinated with the quadrivalent Gardasil, it would have not protected her from her multiple HPV-associated anogenital cancers because her HPV types are not covered by Gardasil-4. Therefore, the overall benefit of the use of Gardasil-4 in the country has to be investigated further.

### 4.2. Precancerous Lesions of the Cervix

The prevalence of precancerous lesions of the cervix or VIA/VILI-DC lesions among our FSW population of 9.9% was similar to what we have found in our general population [38]. One would have expected a higher prevalence of VIA/VILI-DC lesions given the high prevalence of HPV in this population. However, since HPV types 16 and 18 are the most virulent types, causing over 70% of pre-invasive cervical cancer/cancer cases worldwide, the low prevalence of these virulent types might be a protection against high rates of invasive cervical cancers in this population. In Uganda, the prevalence of positive VIA among FSWs was as low as 6% [39]. However, in the Ugandan study, the VIA was performed using the naked eye compared to our study where VIA was enhanced using DC. As reported in the article, it could also be argued that the midwives who performed the VIA in the Ugandan study were newly trained in VIA, whereas our nurses had been doing VIA/VILI-DC for several years and were certainly more experienced. In a study in India in which FSWs were screened with VIA, VILI, and cytology, the prevalence of VIA was 15.2%, that of VILI was 12.2%, and that for cytology was 22.7%, but the overall prevalence of CIN 2–3 was as low as 4.7% [40]. It is not understood why, despite the very high HPV prevalence among FSWs, their prevalence of precancerous lesions of the cervix remains very low.

Both in our study in Cameroon and the meta-analysis of Farahmad et al., HPV type 53 occupied the third and second positions, respectively, in terms of prevalence. This suggests that HPV type 53 has a high prevalence among FSWs in SSA. The oncogenicity of HPV type 53 has been highly questioned. A large multi-country observational study did not find any carcinogenicity with HPV type 53 [41]. This could be one of the contributory factors for the low prevalence of cervical precancer/cancer lesions among our FSW population despite their high HPV prevalence. However, it will also be important to compare the rates of type 53 among women in the general population to understand if there is a particular affiliation between type 53 and FSWs or not.

### 4.3. Treatment Uptake

We used the HPV see-and-treat approach where every participant positive for HPV was provided treatment with TA if they did not have a VIA/VILI-DC lesion. However, if they had lesions on VIA/VILI-DC, the treatment depended on the characteristics of the lesion [13]. As noted in Figure 1, a total of 164 FSWs were treated using TA while 10 were treated using LLETZ. It will be important to do a long-term follow-up among these FSWs who receive treatment to evaluate posttreatment HPV clearance and persistence.

HIV status and site of screening were the only variables in our study that were significantly associated with treatment uptake in the bivariate model but these significances were lost in the multivariable model. It is expected that persons living with HIV are generally likely to be more conscious of their health situations than those who are HIV-negative, due to frequent hospital appointments.

The monthly income generated was not significantly associated with treatment uptake. Treatment with TA and LLETZ requires a mandatory one-month period of abstinence and this was a major concern for FSWs because they were going to stay a month without generating income. This signifies that FSWs may have issues related to health care access and usage. However, this was not investigated in our study. Future studies are necessary to investigate access to health care, monitoring, or programs available to sex workers in the country.

One surprising finding from our study was the number of FSWs who declined to respond to the smoking question. Of the 599 FSWs, 548 (91.5%) indicated that they did not want to respond to the smoking question. The question on the different types of sexual practices appeared to be the most sensitive in our study. However, only 9 (1.5%) FSWs declined to respond to the question on sexual practices. This is an indication that smoking was the most sensitive thing among the FSWs in our cohort. In a Brazilian study to assess the prevalence of smoking among FSWs, 71.5% of participants admitted smoking [42]. However, that study was limited by a small sample size of 83 participants and the proportion of FSWs who declined to participate in the study was not mentioned. The reason why the FSWs in our study declined to respond to the question on smoking is not known. A qualitative study is needed to explore these reasons.

### 4.4. Strengths and Weaknesses

Our study had several strengths. It is one of the first studies to investigate the prevalence of high-risk HPVs and HPV-associated lesions of the cervix among FSWs in Cameroon. Our sample size is one of the largest among all studies in SSA that have studied HPVs among FSWs. We collaborated with an organization (HF) that has worked with FSWs for several years, building the trust needed to enhance data accuracy. All demographic data were collected by HF staff who were versed with the FSWs and their way of life. Compared to other studies in SSA that recruited FSWs as young as 15 and 18 years, our study recruited from the age of 30 years. Therefore, we worked with an age range that is more likely to have persistent rather than transient HPV infection. Finally, we used an HPV see-and-treat approach, offering treatment to all FSWs who were positive for high-risk HPVs, whereas, previous studies did not report or perform treatment. We acknowledge a few weaknesses in our study. (1) The sample size was not evenly distributed among the three study sites. (2) There was no real test to verify if all participants enrolled were actually sex workers because any woman who self-identified herself as a FSW was automatically considered to be one. It is possible that a few non-FSWs could have identified themselves as FSWs just to benefit from the project. In this case, a bias could have been created towards the null.

## 5. Conclusions

The prevalence of HPV infection among FSWs in Cameroon is 62.1% which is higher than the worldwide pooled prevalence of FSWs of 42.6%. The most prevalent oncogenic HPV types among FSWs in Cameroon were types 51, 53, and 52, while the least prevalent types were types 18 and 16. Further investigation is needed to explore why types 16 and 18 are the least prevalent in Cameroon, whereas they are the most prevalent in other settings. The main variable associated with HPV infection and treatment uptake among FSWs in Cameroon was positive HIV status. Despite the high HPV prevalence, the prevalence of precancerous lesions of the cervix among FSWs was similar to that of the general population. Further studies are required to understand the reasons for the low prevalence of precancerous lesions of the cervix among FSWs despite their very high prevalence of oncogenic HPVs as well as the reasons for lower uptake of treatment among those with precancerous lesions compared to those without. The smoking question was the most sensitive among all the questions asked to the FSWs as 91.5% declined responding to the question. There is probably something behind smoking among the FSWs that needs to be uncovered in future studies.

## Figures and Tables

**Figure 1 cancers-16-00243-f001:**
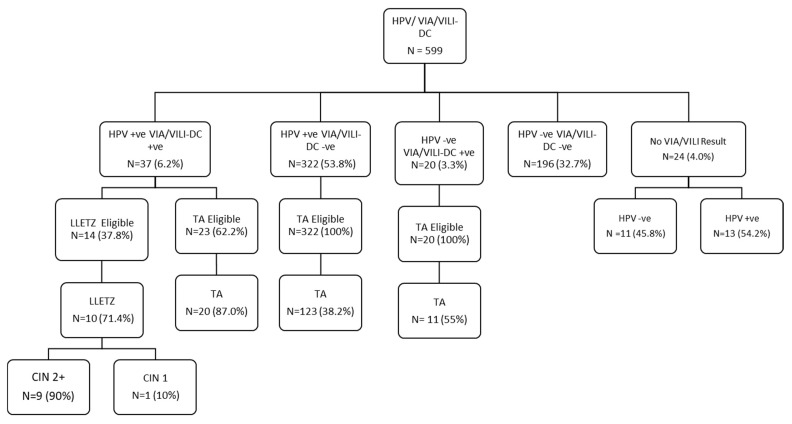
Project algorithm. CIN = cervical intraepithelial neoplasm; HPV = human papillomavirus; VIA/VILI-DC = visual inspection with acetic acid/visual inspection with Lugol’s iodine-digital cervicography; LLETZ = large loop excision of the transformation zone; TA = thermal ablation; −ve = negative; +ve = positive.

**Figure 2 cancers-16-00243-f002:**
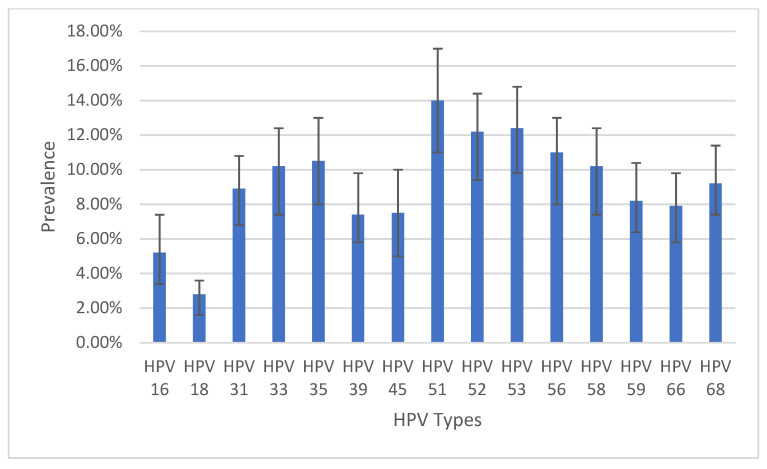
Prevalence of infection by HPV types with 95% confidence intervals. N.B.: Patients could be infected with multiple HPV types.

**Table 1 cancers-16-00243-t001:** Participants’ characteristics for women ≥ 30 according to HPV Results *(N* = 599).

Characteristics	HPV Negative *N* = 227 (37.9%) 95 %CI: (34.0–41.8%)	HPV Positive *N* = 372 (62.1%) 95 %CI: (58.2–66.0%)	*p* Value
Age 30–39 40–49 ≥50	124 (36.3) 83 (40.7) 20 (37.7)	218 (63.7) 121 (59.3) 33 (62.3)	0.587
Marital Status Married Single Others	16 (44.4) 174 (36.2) 32 (43.8)	20 (55.6) 307 (63.8) 41 (56.2)	0.310
Educational Level 0–7 years 8–14 years 15–17 years ≥18	46 (34.8) 98 (40.5) 10 (29.4) 7 (58.3)	86 (65.2) 144 (59.5) 24 (70.6) 5 (41.7)	0.228
Site of Screening Bafoussam Yaoundé Douala	17 (53.1) 97 (45.8) 113 (31.8)	15 (46.9) 115 (54.2) 242 (68.2)	0.001
Other Occupation Domestic worker/housewife Trader/farmer Student Others	191 (37.8) 11 (35.5) 5 (45.5) 12 (54.6)	314 (62.2) 20 (64.5) 6 (54.5) 10 (45.4)	0.418
HIV Status Positive Negative	158 (42.0) 51 (28.3)	218 (58.0) 129 (71.7)	0.002
Religion Baptist Catholic Presbyterian Pentecostal Muslims Others	16 (35.6) 117 (36.6) 36 (40.0) 23 (37.7) 8 (53.3) 17 (38.6)	29 (64.4) 203 (63.4) 55 (60.4) 38 (62.3) 7 (46.7) 27 (61.4)	0.852
Duration in Sex work <3 years 3–7 years ≥7 years	52 (40.3) 99 (34.6) 59 (38.8)	77 (59.7) 187 (65.4) 93 (61.2)	0.468
Number of clients per day ≤5 6 to 10 11–14 ≥15 Do not want to respond	126 (35.4) 63 (38.0) 4 (44.4) 10 (47.6) 24 (51.1)	230 (64.6) 103 (62.0) 5 (55.6) 11 (52.4) 23 (48.9)	0.247
Income per month XAF < 50,000 XAF ≥ 50,000 Do not want to respond	34 (31.5) 118 (34.4) 75 (50.7)	74 (68.5) 225 (65.6) 73 (49.3)	0.001
Condom use Always Sometimes Depends on the customer’s preference Never Do not want to answer	138 (36.2) 56 (35.0) 23 (53.5) 1 (50.0) 9 (75.0)	243 (63.8) 104 (65.0) 20 (46.5) 1 (50.0) 3 (25.0)	0.009
Alcohol per week 0 to 5 bottles 6 to 10 bottles 11 to 14 bottles ≥15 bottles Do not want to respond	73 (34.3) 47 (38.2) 21 (43.8) 29 (40.9) 57 (39.6)	140 (65.7) 76(61.8) 27 (56.2) 42 (59.1) 87 (23.4)	0.676
Smoking per week 0 to 14 ≥15 Do not want to respond	10 (41.7) 10 (37.0) 207 (37.8)	14 (58.3) 17 (63.0) 341 (62.2)	0.925
Responded to smoking question Do not want to respond Responded	207 (37.8) 20 (39.2)	341 (62.2) 31 (60.8)	0.839
* Other sexual practices Oral sex Anal sex Sexting Group sex Do not want to answer	22 (34.4) 10 (37.0) 1 (20.0) 16 (30.8) 4 (44.4)	42 (65.6) 17 (63.0) 4 (80.0) 36 (69.2) 5 (55.6)	0.539 0.925 0.656 0.268 0.736
Treatment uptake Yes	0 (0.0)	141 (100.0)	<0.0001

* One could conduct multiple sex activities, so the column does not add up to 100%. Instead, each activity adds up to 100%.

**Table 2 cancers-16-00243-t002:** Logistic regression on predictors for infection with HPV (*N* = 556 included in the adjusted model).

Predictors	HPV Infection					
	cOR	95% CI	*p*-Value	aOR	95% CI	*p*-Value
Age 30–39 40–49 ≥50	Ref 0.83 0.94	-- 0.58–1.18 0.52–1.71	-- 0.302 0.835			
Marital Status Married Single Others	Ref 1.41 1.03	-- 0.71–2.80 0.46–2.29	-- 0.323 0.952			
Educational Level 0–7 years 8–14 years 15–17 years ≥18	2.62 2.06 3.36 Ref	0.79–8.71 0.64–6.67 0.86–13.14 --	0.117 0.229 0.082 --			
Site of Screening Bafoussam Yaoundé Douala	Ref 1.34 2.43	-- 0.64–2.83 1.17–5.03	-- 0.437 0.017	Ref 1.49 2.18	-- 0.64–3.46 0.93–5.09	-- 0.351 0.072
Other Occupation Domestic worker/housewife Trader/farmer Student Others	Ref 1.11 0.73 0.51	-- 0.52–2.36 0.22–2.42 0.22–1.20	-- 0.794 0.607 0.121			
HIV Status Positive Negative	1.83 Ref	1.25–2.69 --	0.002 --	1.65 Ref	1.11–2.45 --	0.014 --
Religion Baptist Catholic Presbyterian Pentecostal Muslim Others	1.10 1.05 0.93 Ref 0.53 0.96	0.49–2.44 0.60–1.85 0.48–1.80 -- 0.17–1.65 0.43–2.14	0.821 0.865 0.818 -- 0.274 0.923			
Duration in Sex work <3 years 3–7 years ≥7 years	Ref 1.28 1.06	-- 0.83–1.96 0.66–1.72	-- 0.265 0.799			
Income per month XAF < 50,000 XAF ≥ 50,000 Do not want to respond	Ref 0.88 0.45	-- 0.55–1.39 0.27–0.75	-- 0.576 0.002	Ref 0.75 0.49	-- 0.45–1.24 0.27–0.89	-- 0.261 0.019
Condom use Always Sometimes Depends on the customer’s preference Never Do not want to answer	Ref 1.06 0.49 0.57 0.19	-- 072–1.55 0.26–0.93 0.04–9.15 0.05–0.71	-- 0.787 0.029 0.690 0.014	Ref 1.38 0.50 1.26 0.50	-- 0.89–2.15 0.25–1.00 0.08–20.90 0.12–2.05	-- 0.152 0.049 0.874 0.337

N.B.: The adjusted model includes the site of screening, HIV status, condom use, and income groups. aOR = adjusted odds ratios, cOR = crude odds ratios.

**Table 3 cancers-16-00243-t003:** Participants’ characteristics according to VIA/VILI lesions (*N* =575).

Characteristics	VIA/VILI Negative *N* = 518 90.1%	VIA/VILI Positive *N* = 57 9.9%	*p* Value
Age 30–39 40–49 ≥50	291 (88.2) 177 (92.2) 50 (94.3)	39 (11.8) 15 (7.8) 3 (5.7)	0.186
Marital Status Married Single Others	33 (91.7) 411 (89.2) 67 (95.7)	3 (8.3) 50 (10.8) 3 (4.3)	0.219
Educational Level 0–7 years 8–14 years 15–17 years ≥18	119 (91.5) 204 (89.5) 27 (84.4) 10 (90.9)	11 (8.5) 24 (10.5) 5 (15.6) 1 (9.1)	0.607
Site of Screening Bafoussam Yaoundé Douala	27 (90.0) 188 (94.5) 303 (87.6)	3 (10.0) 11 (5.5) 43 (12.4)	0.035
Other Occupation Domestic worker/housewife Trader/farmer Student Others	433 (89.3) 29 (93.5) 9 (100.0) 19 (86.4)	52 (10.7) 2 (6.5) 0 (0) 3 (5.3)	0.728
HIV Status Positive Negative	150 (85.7) 331 (91.7)	25 (14.3) 30 (8.3)	0.033
Religion (*N* = 555) Baptist Catholic Presbyterian Pentecostal Muslim Others	39 (90.7) 277 (90.8) 80 (90.9) 50 (82.0) 13 (86.7) 39 (90.7)	4 (9.3) 28 (9.2) 8 (9.1) 11 (18.0) 2 (13.3) 4 (9.3)	0.441
Duration in Sex work <3 years 3–7 years ≥7 years	114 (90.5) 243 (88.7) 134 (92.4)	12 (9.5) 31 (11.3) 11 (7.6)	0.506
Number of clients per day ≤5 6 to 10 11–14 ≥15 Do not want to respond	303 (88.6) 147 (91.3) 8 (47.1) 19 (100.0) 41 (91.1)	39 11.4() 14 (8.7) 9 (51.9) 0 (0) 4 (8.9)	0.530
Income per month XAF < 50,000 XAF ≥ 50,000 Do not want to respond	99 (93.4) 293 (88.5) 126 (91.3)	7 (6.6) 38 (11.5) 12 (8.7)	0.295
Condom use Always Sometimes Depends on the customer’s preference Never Do not want to answer	330 (90.2) 142 (92.2) 34 (82.9) 2 (100.0) 10 (90.9)	36 (9.8) 12 (7.8) 7 (17.1) 0 (0) 1 (9.1)	0.449
Alcohol per week 0 to 5 6 to 10 11 to 14 ≥15 Do not want to respond	192 (92.3) 111 (92.5) 40 (83.3) 53 (85.5) 122 (89.0)	16 (7.7) 9 (7.5) 8 (16.7) 9 (14.5) 15 (11.0)	0.198
Smoking per week 0 to 14 ≥15 Do not want to respond	20 (95.2) 23 (92.0) 475 (89.8)	1 (4.8) 2 (8.0) 54 (10.2))	0.922
Responded to smoking question Do not want to respond Responded	475 (89.8) 43 (93.5)	54 (10.2) 3 (6.5)	0.607
* Other sexual practices Oral sex Anal sex Sexting Group sex Do not want to answer	55 (91.7) 23 (88.5) 5 (100.0) 44 (89.8) 7 (77.8)	5 (8.3) 3 (11.5) 0 (0.0) 5 (10.2) 2 (22.2)	0.665 0.735 1.000 1.000 0.222

* One could conduct multiple sex activities, so the column does not add up to 100%. Instead, each activity adds up to 100%.

**Table 4 cancers-16-00243-t004:** Participants’ characteristics according to treatment uptake.

	Not Treated *N* 240 (63.3)	Treated *N* = 139 (36.7)	*p* Value
VIA/VILI Positive Negative	30 (52.6) 210 (65.2)	27 (47.4) 112 (34.8)	0.069
Age 30–39 40–49 ≥50	151 (66.8) 68 (57.1) 21 (61.8)	75 (33.2) 51 (42.9) 13 (38.2)	0.204
Educational Level 0–7 years 8–14 years 15–17 years ≥18	62 (70.5) 90 (62.5) 17 (68.0) 3 (50.0)	26 (29.5) 54 (37.5) 8 (32.0) 3 (50.0)	0.507
HIV Status Positive Negative	74 (56.1) 150 (67.6)	58 (43.9) 72 (32.4)	0.030
Site of Screening Bafoussam Yaoundé Douala	12 (85.7) 84 (75.7) 144 (57.4)	2 (14.3) 30 (26.3) 107 (42.6)	0.002
Number of clients per day ≤5 6 to 10 11–14 ≥15 Do not want to respond	137 (58.6) 72 (67.9) 3 (60.0) 8 (88.9) 20 (80.0)	97 (41.4) 34 (32.1) 2 (40.0) 1 (11.1) 5 (20.0)	0.060
Income per month XAF < 50,000 XAF ≥ 50,000 Do not want to respond	49 (64.0) 138 (57.5) 54 (73.0)	27 (36.0) 92 (40.0) 20 (27.0)	0.130
Alcohol per week 0 to 5 6 to 10 11 to 14 ≥15 Do not want to respond	86 (60.6) 55 (69.6) 22 (68.8) 26 (66.7) 51 (58.2)	56 (39.4) 24 (30.4) 10 (31.2) 13 (33.3) 36 (25.9)	0.519
Smoking per week 0 to 14 ≥15 Do not want to respond	6 (50.0) 11 (68.8) 223 (63.5)	6 (50.0) 5 (31.2) 128 (36.5)	0.569
Responded to smoking question Do not want to respond Responded	223 (63.5) 17 (60.7)	128 (36.5) 11 (39.3)	0.766

**Table 5 cancers-16-00243-t005:** Logistic regression on predictors for treatment uptake.

Predictors	Treatment Uptake					
	cOR	95% CI	*p*-Value	aOR	95% CI	*p*-Value
Age 30–39 40–49 ≥50	Ref 1.51 1.25	Ref 0.96–2.38 0.59–2.63	-- 0.077 0.562			
Marital Status Married Single Others	Ref 0.984 0.75	Ref 0.35–2.01 0.25–2.22	-- 0.688 0.604			
Educational Level 0–7 years 8–14 years 15–17 years ≥18	0.42 0.60 0.47 Ref	0.08–2.22 0.12–3.08 0.08–2.87 --	0.306 0.540 0.414 --			
Site of Screening Bafoussam Yaoundé Douala	Ref 2.14 4.46	-- 0.45–10.13 0.98–20.33	-- 0.337 0.054	Ref 1.64 3.30	-- 0.33–8.08 0.69–15.68	-- 0.547 0.134
HIV Status Positive Negative	1.63 Ref	1.05–2.55 --	0.030 --	1.53 --	0.97–2.40 --	0.065 --
Duration in Sex work <3 years 3–7 years ≥7 years	Ref 1.14 1.06	-- 0.66–1.98 0.56–1.99	-- 0.348 0.771			
Income per month XAF < 50,000 XAF ≥ 50,000 Do not want to respond	Ref 1.19 0.66	-- 0.69–2.03 0.33–1.32	-- 0.538 0.240			
Condom use Always Sometimes Depends on the customer’s preference Never Do not want to answer	Ref 0.79 1.38 <0.001 0.54	-- 0.48–1.27 0.50–3.20 <0.001–>999.9 0.06–5.29	-- 0.314 0.457 0.987 0.599			
VIA/VILI Positive Negative	1.69 Ref	0.96–2.98 --	0.071 --			

aOR = adjusted odds ratios, cOR = crude odds ratios, The multivariable model includes HIV status and site of screening.

## Data Availability

The data presented in this study are available in this article.

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
