# Peer review of "Human Papillomavirus Types and Cervical Cancer Screening among Female Sex Workers in Cameroon"

_cancers, 2024, doi:10.3390/cancers16020243_

Round 1

Reviewer 1 Report (Previous Reviewer 2)

Comments and Suggestions for Authors

Dear Authors,

This is a resubmitted version of your original manuscript. You have corrected it according to the suggestions and it can be accepted without any changes now. 

Best regards and good luck

Author Response

Thanks.

Reviewer 2 Report (Previous Reviewer 3)

Comments and Suggestions for Authors

The paper has improved a lot and in this version it can be published. Check the tables. Great job.

Author Response

Thanks.

Reviewer 3 Report (Previous Reviewer 4)

Comments and Suggestions for Authors

Thank you for going through the manuscript and the reviewers' points
In my honest opinion, the authors have responded satisfactorily to the reviewers’ criticisms.

Author Response

thanks

This manuscript is a resubmission of an earlier submission. The following is a list of the peer review reports and author responses from that submission.

Round 1

Reviewer 1 Report

Comments and Suggestions for Authors

In general terms, I find the article very interesting, however I consider that the introduction and methodology could be better written.

Although the WHO recommends that screening of women in the general population should start at the age of 30, this population has special characteristics, such as higher exposure rates.

Account should be taken of the gaps observed in the different methodologies that can be used, which could lead to over-reporting or under-reporting of cases; in this sense, it would have been important to know the data of the population under 30.

A longer follow-up of the treated patients would be of great relevance in order to have a better picture.

With regard to access to health care, monitoring or programmes available to sex workers in the country, did they exist? It's worth mentioning it.

An interesting aspect is the genotypes found, it would be of great value to broaden the discussion in this article, the difference with other studies in which genotypes 16 and 18 are still prevalent and include more information in similar population and of course as mentioned correlate it with the impact of the vaccine. The discussion of the sociodemographic characteristics of the patients included in the study should be expanded. 

The correlation of the lesion found with the genotype and treatment should be included. 

I consider that the work has relevant data, but it should be written in a better way and the most important findings should be interpreted in a broader way, so that it is more attractive to the reader.

Reviewer 2 Report

Comments and Suggestions for Authors

Dear Authors,

This paper addresses a significant topic, however, I would recommend several modifications before considering its publication. Below these are some suggestions for You:

Introduction:

A. The introduction is very informative, but I suggest including some additional information about HPV, like:  

1. It has been established that persistent infection with HPV is also associated with anogenital as well as head and neck cancers [according to your Table 1 Question ‘other sexual practices’ it seems quite important]; 

2. the viral proteins gp120 and tat and the HIV-associated activation of inflammatory processes in the mucosal epithelium, can lead to the disruption of epithelial junctions, which may promote the penetration and/or dissemination of other viruses, including herpesviruses (HSV) and HPV, 

3. HPVs are subdivided into two groups: low-risk (e.g. HPV 1, 2, 6, 8, 11, 34, 40, 42, 43, 44, 61, 69, 71, 72, 81, 83, and 84), commonly associated with benign manifestations (warts, papillomas, condylomata and focal epithelial hyperplasia), and high-risk types (e.g. HPV 16, 18, 31, 33, 35, 39, 45, 51, 52, 53, 56, 58, 59, 66, 68, 73, and 82) associated with malignant manifestations

4. One of the key events of HPV-induced carcinogenesis is the integration of the HPV genome into a host chromosome

https://doi.org/10.3390/ijerph18178999, https://doi.org/10.5114/ada.2021.107269

https://doi.org/10.3390/v15030778

Populations and methods: 

A.Subsection 2.1 should be divided into smaller groups, which help in navigation  

B. Include a producer and country of origin of used detection kits/systems etc. like 'Thermofisher Scientific, Waltham, MA, USA'

Results:

A. Figure 2. Why is HPV-16 not included? 

B. Table 2 and Table 5. I suggest improving presentation 

Discussion: 

A. I suggest explaining HIV infection impact on HPV coinfection

Conclusion

A. The recommendations should inspire future research or medical intervention, not to be a repetition of results

Best regards and good luck

Comments on the Quality of English Language

Minor editing of English language required

Reviewer 3 Report

Comments and Suggestions for Authors

Dear Authors,

The paper is very interesting and important because the target population is very vulnerable.

It is a very unprotected population that needs a lot of health assistance.

Methodologically I consider this paper to be correct. I will only make a few comments

Minor issue

Although 599 women have been selected, which is enough, I would like to know if a study of the sample size was made and why these women were selected.

Table 2 needs to be better ordered for reference.

The abscissa axis of graph 1 is not understood. Please make the legend of the graph clearer.

In table 4, Hairdresser/seamstress4, Jehova witness and erece have an OR that is not understood. Please make it clearer.

Reviewer 4 Report

Comments and Suggestions for Authors

I read with great interest the manuscript, which falls within the aim of this Journal and offers a high-quality overview of the topicIndeed, the topic is of the utmost importance given the efforts made by the WHO for the prevention and early diagnosis of cervical cancer and precancerous cervical lesions. The abstract summarizes sufficiently the contents of the manuscript and the introduction is satisfactory. The tables and figures are clear and interesting.

Although the manuscript can be considered already of high quality, I would suggest taking into account the following minor recommendations:

- I suggest another round of language revision, to correct a few typos and improve readability.

- Inclusion/exclusion criteria should be better clarified by extending their description.

- Discussions can be expanded by citing relevant articles about long-term risk factors for the recurrence of HVP-related lesions (see PMID: 37401466).

- I also suggest authors to better organize the discussion section following this ideal structure: main findings of the study, strengths and limitations of the study, implications and comparison with literature and future directions. 

-What are the implications of these findings for clinical practice and/or further research?

-What are the strengths and limitations of this manuscript?

Comments on the Quality of English Language

Minor editing of the English language is required to make the work clearer and more readable.